# Badminton Activity Recognition Using Accelerometer Data

**DOI:** 10.3390/s20174685

**Published:** 2020-08-19

**Authors:** Tim Steels, Ben Van Herbruggen, Jaron Fontaine, Toon De Pessemier, David Plets, Eli De Poorter

**Affiliations:** 1IDLab, Department of Information Technology, Ghent University-imec, 9000 Ghent, Belgium; tim.steels@ugent.be (T.S.); ben.vanherbruggen@ugent.be (B.V.H.); jaron.fontaine@ugent.be (J.F.); 2WAVES, Department of Information Technology, Ghent University-imec, 9000 Ghent, Belgium; toon.depessemier@ugent.be (T.D.P.); david.plets@ugent.be (D.P.)

**Keywords:** badminton, activity recognition, accelerometer, gyroscope, DNN, CNN, neural network, machine learning

## Abstract

A thorough analysis of sports is becoming increasingly important during the training process of badminton players at both the recreational and professional level. Nowadays, game situations are usually filmed and reviewed afterwards in order to analyze the game situation, but these video set-ups tend to be difficult to analyze, expensive, and intrusive to set up. In contrast, we classified badminton movements using off-the-shelf accelerometer and gyroscope data. To this end, we organized a data capturing campaign and designed a novel neural network using different frame sizes as input. This paper shows that with only accelerometer data, our novel convolutional neural network is able to distinguish nine activities with 86% precision when using a sampling frequency of 50 Hz. Adding the gyroscope data causes an increase of up to 99% precision, as compared to, respectively, 79% and 88% when using a traditional convolutional neural network. In addition, our paper analyses the impact of different sensor placement options and discusses the impact of different sampling frequenciess of the sensors. As such, our approach provides a low cost solution that is easy to use and can collect useful information for the analysis of a badminton game.

## 1. Introduction

Badminton is an Olympic discipline and it is one of the most popular racket sports worldwide. Both at the recreational and professional level, an analysis of the movements can be of great added value during the training process [1]. Technologies that can help to optimize training are constantly being sought in order to improve the personal performance of badminton players. This includes determining the movements of the player during training and game situations. Badminton is a sport in which tactics, technique, and the precise execution of the movements are of great importance. Players and coaches are interested in affordable and compact sensor hardware that have the ability to capture the necessary measurements to obtain relevant, useful information.

The recognition and monitoring of human activities is a research area that has been thoroughly investigated in recent years [2,3]. The increase in popularity of smart wearable devices, such as smartwatches and smartphones, has facilitated the necessary data collection process. These devices have many built-in sensors, such as an accelerometer and a gyroscope, which are sufficiently accurate to be useful for the recognition of human activities. Activity recognition is very intriguing, because it finds applications in a wide range of domains. Some of the interesting applications are tracking the physical activity of athletes [4], predicting the movement of robots and vehicles using sensors [5], etc. The aim of this work is to recognize badminton strokes based on measured accelerometer and gyroscope data. The obtained signals from the accelerometer are then processed by a computer model. The idea is that the system ’learns’ to distinguish the different strokes, based on (a subset of) these measurements. The aim is to achieve an accurate, automatic classification of the movements based on supervised machine learning. Classifications, predictions, and interpretations can be made more accurately by training the system on more varied data (different players, ...). The purpose of this classification is to gain a better understanding of a player’s strengths and vulnerabilities. Knowledge of the played strokes provides a good insight into the extent to which a player e.g., mainly plays offensive or defensive. The results of these data can be used as support during the training process of both amateur and professional players.

Currently, the evaluation of training and games are mainly done either by manually by the coach or spectators or with video based systems [6]. While the first requires time and attention from the coach which might not be always accessible, the latter requires analyzing time, high installation costs, resolution issues, and/or privacy concerns, the coach perspective is still positive towards performance analysis at an objective manner [7]. To cope with these problems, a small accelerometer device at the racket can provide a more general method, accessible to all levels of players [8].

A big difference with existing technologies is that we go deeper into the specific movements in badminton, where other studies focus more on coarse-grained activity recognition or other sports. In addition, of-the-shelf sensors are used during this project. We also investigate how we can classify movements with a long and short duration as correctly as possible without using a manually chosen fixed frame size. A frame is an input vector for the machine learning model. The techniques used in this research are not only limited to badminton, but are useful in different types of activity recognition.

The main contributions of this paper are as follows:We propose a novel activity recognition approach based on ensemble learning using accelerometer and gyroscope data from different frame sizes.We classify nine different badminton strokes/movements.We investigate the influence of the sampling frequency on the accuracy of the model.We investigate the impact of the sensor position on the classification accuracy.The used data set of nearly two hours has been made open source.

The remainder of this paper is organized as follows. Section 2 discusses related work and the most modern activity recognition techniques. Section 3 describes the measuring techniques for collecting the data, the data set, and the used sensors. Section 4 proposes the novel classification approach. This includes pre-processing the data and a description of the used model. Section 5 discusses the obtained results. Finally, in the last two sections, the conclusions are formed and future work is discussed.

## 2. Related Work

As mentioned, quite a lot of research has already been carried out in the domain of activity recognition. Mobile devices have become very popular. These devices include an accelerometer, gyroscope, and magnetometer that are sufficiently accurate to recognize human daily activities (e.g., walking, running, sitting, taking stairs, ...) [9,10,11,12,13,14]. This research area is intriguing, because it finds applications in a wide range of domains. Hence, this technology is very often used in everyday life without knowing it. A wide variety of methods for monitoring activities have appeared in last decades. For various sports, extensive research has already been done into recognizing the movements with various techniques. Existing work usually only focuses on course-grained activities or they distinguish different sport categories. In-depth research that specifically focuses on a user friendly way to monitor badminton is non-existing (to the authors’ knowledge). Tennis is one of the sports that is most similar to badminton in terms of movements and strokes. The strokes are performed differently, but the concepts in activity recognition are similar. Thus, one could train new models that are specified for new movements in the same way. More in-depth knowledge about the specific sport/movements is required in order to further optimize the results. These technologies often have a high cost and systems often need to be calibrated. Not all options are equally easy to use and user friendly. To develop a system that is valuable in both professional environments and commercial applications, these things must be taken into account. In Table 1, we give a short overview of related previous research. There are two main categories: solutions based on accelerometer and gyroscope and solutions based on computer vision.

However, the studies mentioned do not always go deeper into a wide variety of movements. In [15], coarse-grained activities (walking, running, lying, etc.) are recognized and, in [16], only the difference between a service and a non-service is investigated. The number of recognized strokes is often limited and the focus is mainly on finding just a working solution without further analyzing which factors influence the performance. Reference [8] is quite similar in terms of strokes and approach to our research, but here the chosen position of the sensors is less user friendly and only five activities are recognized. Due to the use of a novel neural network architecture (ensemble learning based on different timeframes), our work also achieves a higher accuracy (99% vs. 88.9%). In [17], a convolutional neural network delivers accuracies of up to 96.5% for their classification of tennis strokes. Finally, Reference [18] also uses a CNN, but this time by using visual recognition, achieving an accuracy around 98.7%.

In contrast to the mentioned papers, our paper analyzes, in depth, the performance trade-offs of different system design parameters. For example, we investigate the minimum sampling frequency and the best position to place the sensors on the body. The impact on accuracy when fewer sensor axes are used will be examined. We also have a look at the extent to which CNNs make better predictions than Deep Neural Networks (DNNs) without convolutional layers.

## 3. Data Collection

### 3.1. Measurements

A fairly extensive data set of about two hours was captured at the beginning of the study. The badminton strokes for two right-handed test persons were measured. These subjects, a male and a female player, are 21 years old and practice badminton at an amateur level. Each was asked to play the different strokes consecutively as often as possible. Subsequently, these persons were also asked to play a varied scenario, in which all kinds of strokes occurred. Movements were recorded while using an accelerometer and gyroscope, which were attached to the human body or on the badminton racket. The shuttle was not hit during these measurements to make sure the stroke can be executed in a controllable environment for training This way, the model does not depend on shuttle speed/position or players movement. The data are measured at 100 Hz and for evaluating purposes, we subsample the dataset immediately after collection to 50, 25 and 12.5 Hz. The model is evaluated using these different sampling frequencies to deduce the minimum frequency that is required for activity recognition. The advantages of a lower sampling frequency are the lower required storage capacity and the longer battery life. The total data set that was measured consists of 663,954 samples that each contain data from six axes resulting in 6639 actions with a frame size of 1 s and sampling rate of 100 Hz.

### 3.2. Defining Activities

Seven important strokes were selected for classification:**Overhead Defensive Clear** (overhand-forehand): the player hits the shuttle overhead from the backfield upwards to the opponent’s rear court to get back into position for the next strike of the opponent in time.**Dab** (overhand-forehand): the shuttle is struck steeply and quickly downwards from the front court on the opponent’s ground.**Drive** (underhand-forehand): the shuttle is hit in a quick, underhand motion.**Short serve** (underhand-backhand): this is often the first trick played in a game situation.**Lob** (underhand-forehand): the shuttle is hit high and deep in the rear court with an underhand stroke with the aim of getting the opponent as far as possible in the back of the court.**Net drop** (overhand-backhand): the netdrop is a short ball over the badminton net. However, the shuttle departs from the front of the field.**Smash** (overhand-forehand): the shuttle is hit offensive and downwards with an overhand strike, so that the opponent has to play a defensive shot.

We show this in Figure 1. In addition, we also want to be able to classify running movements and moments of standstill.

### 3.3. Sensors

The device used during the experiments in this work is the Axivity AX6 6-Axis Logging Device [19,20]. Both the dimensions (23 mm × 32.5 mm × 8.9 mm) and weight (11 g) permit attachment to the racket without disturbance to the player. The AX6 features a six-axis motion sensor that measures linear acceleration and angular velocity. Accordingly, it includes both an accelerometer and a gyroscope. In addition, the device also includes the temperature sensor, a light sensor for measuring ambient light and a real-time clock. In this paper, only the accelerometer and gyroscope are used. The accelerometer signal is truncated at ±16 g, the gyroscope at ±2000 dps. All of the collected data are stored in binary format in the built-in flash memory. To allow further processing of the results, the recorded CWA files must be manually transferred to a computer device and converted to CSV files with the OMGUI software tool [21].

### 3.4. Variety of Sensors

The combinations of sensor axes that we will investigate are only accelerometer data or a combination of both accelerometer data and gyroscope data. The disadvantage of the gyroscope is the typical high power consumption, resulting in bigger batteries or shorter lifetime. Moreover, extra data storage will be needed. Because the Axivity AX6 sensor is used, we must take into account the internal memory storage that is not infinite. When we do not store the data locally but process it in real-time, we still have to take power consumption into account. Turning on the gyroscope drastically reduces the battery life of a mobile device.

### 3.5. Position of Sensors

We obtain other information that is measured depending on the position where the sensors are placed. The challenge lies in finding the most optimal location for measuring the necessary data for classification of the strokes. During different experiments, different locations for attaching the sensors are considered: the bottom of the racket’s grip, the wrist, and the upper arm. We show this in Figure 2.

In addition, ease of use was taken into account when choosing the positions. Placing a sensor in the middle of the strings of the racket is not very practical in a real badminton game. These three positions were determined in consultation with two experts in the field of badminton. A brief survey of four people with badminton-experience shows that players accept the sensor on the bottom of the racket’s grip.

## 4. Classification Process

Once the data have been captured, it must be converted into a way that is easy to interpret for sports analysts and trainers. In the end, the player and his team should have easy access to his game statistics: the strokes he played, how many strokes he played, when he played this strokes, etc.

### 4.1. Signal Preprocessing

For deep learning itself, the libraries TensorFlow [22] and Scikit-learn [23] are used. A large amount of data is given as input during the training process in order for the computer model to recognize activities. Based on these data, the model learns the typical course of linear and angular accelerations for each stroke. The input is taken as varied as possible in order to obtain a generically applicable model that produces good results for different people. The execution of a stroke is different for everyone. The data consist of series of the same successive strokes as well as realistic scenarios with alternating movements and displacements between. In addition, all of the samples are labeled according to the activity to which they belong.

The input data for the model are first scaled, but no additional signal processing is done. The training samples are recorded without idle time between the strokes to retrieve ground truth. For moments where the player is not playing, we added an extra scenario to make sure inherent sensor noise is not classified as stroke. Unlike many other previous studies [24], we do not normalize our samples, because, in addition to the learning of the patterns in the signal, magnitudes also possess information for the model to learn. Subsequently, we divide the signals into frames of fixed time duration. Because a large number of badminton strokes involve similar movement patterns, it is important to consider time frames that contain sufficiently large parts of a stroke at once during the classification. This allows for the model to learn to recognize the full course of a specific stroke movement. For each frame, the most common ground truth label is used as the label for the entire frame. We do not use overlapping frames during the training process.

### 4.2. Layers Cnn

For implementing a Convolutional Neural Network, Keras’ Conv2D class is suitable for our goal. The Sequential model type is, in this case, the easiest way to build a model and allows layer by layer adaptation. The first layer that we add to our network is a Conv2D layer. We give the first parameter, “filters”, a value of 16. The parameter “kernelsize” is the size of the filter matrix for our convolution. The kernel size chosen means that we will have a 3 × 3 filter matrix. Activation is the activation function for the layer. The function that we use for our first two layers is the ReLU or Rectified Linear Activation. This activation function has been proven to work well in neural networks [25]. When data pass through a deep neural network, the values change, making some too large or too small. By normalizing the data per batch, any disadvantages that are associated with this are filtered out. This usually ensures better end results.

Dropout layers are added several times. Dropout is a technique in which randomly selected neurons are ignored during training. This prevents overfitting of the model. A “Flatten” layer is added between the Conv2D layers and the Dense layer. Flatten acts as a connection between the convolution and Dense layers. “Dense” is the layer type that we will use for our output layer. Dense is a standard layer type that is used in many cases for neural networks.

We have nine nodes in our output layer, one for every possible outcome. Seven labels are the different strokes that we detect. We stick the other two labels on the displacements of the player and moments of rest. The activation is “softmax”. Softmax makes the output sum to 1, so that the output can be interpreted as probabilities. The model then makes its predictions based on which option has the greatest probability. Table 2 provides a brief overview of the different layers and their output dimensions. Figure 3 gives a brief overview.

Overall, the complexity of this model is low, with 316,809 trainable parameters for a frame size of 800 ms. Moreover, a high end desktop (Intel Core i9 9900k, NVIDIA TITAN RTX (Intel and NVIDIA corperation, California, US)) achieves a classification rate of 20,953 samples/second, with an average duration of 0.0477 ms per classification. We have deployed the model on a NVIDIA Jetson Nano to illustrate the feasibility of running the proposed solution on embedded hardware. Here, the Jetson achieves a classification rate of 3515 samples/second, with an average duration of 0.284 ms per classification. The achieved performance indicates low complexity of the proposed solution and high feasibility for embedded implementation.

### 4.3. Remove Impurities in Predictions

The classification by the model is not flawless. For example, when switching from a specific striking movement to moments of rest or displacement, the classification can go wrong. As already mentioned, the most common label in a frame is used as the label for the entire frame and the model uses Softmax, which is based on probabilities. We notice that, with misclassifications, the model is quite uncertain about its prediction.

## 5. Results and Evaluation

The evaluation is done by training the model on the data of the male player (series of the same successive strokes as well as scenarios) and by predicting a scenario of the unseen, female player. These scenarios contain every type of stroke, interspersed with rest and running moments. Table 3 shows the number of times a type of strokes is played in the scenario (test data). The number of strokes in the training data is also displayed.

### 5.1. Simple Accelerometer Based Model

Our first solution classifies the strokes using only the accelerometer data. Figure 4 shows the results of using accelerometer data sampled at 12.5 Hz. Over all classes, the weighted average accuracy is 76%. Rest and displacement are always recognized with very high accuracy, regardless of frame size, sampling frequency, and number of sensors. This can be explained, since the signal of these movements is not very similar to any other class. Therefore, they are very easy to distinguish. What is most striking is the fact that almost all drives (98%) are classified as a clear. Additionally, the dab is not properly recognized and there is confusion between the smash and the clear.

Increasing the sampling frequency has a limited influence on the predictions. At 25 Hz, the weighted average accuracy is 77%, at 50 Hz 79%, and finally 82% at a sampling frequency of 100 Hz. Similar to before, we still observe misclassifications of the drive when no gyroscope data are used. However, at very higher sampling frequencies (starting from 100 Hz), the differences between these two strokes (drive and clear) starts being observable for our model: at 100 Hz, the drives are recognized 48% of the time using only accelerometer data. Better results will be obtained later after adding the gyroscope data.

### 5.2. Fixed Frame Size Versus Variable Frame Size

Different frame sizes (time durations) will be investigated to further improve the accuracy. Depending on the frame size used for training, the accuracy for the different types of strokes varies strongly. Strokes with a short duration are more accurately predicted when the frame size is smaller. For strokes that have a longer duration, the recognition is better when using larger frame sizes. Table 4 shows the optimal frame sizes per stroke type.

To exploit this fact, we utilize ensemble learning [26,27]. More specifically, we train multiple models each with a different frame size. Frame sizes of 0.4, 0.6, 0.8, 1, 1.2, 1.4, and 1.6 s are used. These time intervals cover the average duration of the different strokes. All other parameters during the training of the model remain unchanged. Table 5 provides an overview of the accuracy per fixed frame size. After the predictions, the output of all these models is combined to arrive at a composite whole. In this way, we try to obtain an optimal combination of the different frame sizes. By combining these different models, we obtain an overall better result: a traditional convolutional neural network with fixed input frames has a weighted average accuracy of 87%, as compared to 98% when combining different models with different input frame lengths. The combined model performs roughly well for every stroke type. Therefore, we use this approach during the rest of this paper.

### 5.3. Added Value Gyroscope Data

We repeat the exact same experiment keeping all other parameters constant in order to compare the accuracy obtained with or without gyroscope data. We investigate the impact at different sample frequencies: 12.5 Hz, 25 Hz, 50 Hz and 100 Hz. We also use ensemble learning, as described in Section 5.2. In Figure 5 three new metrics are evaluated to further indicate the performance of the model in terms of false negatives and false positives: precision, recall, and F1 score. F1 score is defined, as follows:(1)F1=2∗(precision∗recall)/(precision+recall)
where precision=tp/(tp+fp), recall=tp/(tp+fn), tp = true positives, fp = false positives, and fn = false negatives. Figure 5a shows the results that were obtained with only accelerometer data. Figure 5b shows the results when we add the gyroscope data. Omitting the gyroscope increases battery lifetime, but it has a negative influence on the predictions of the model. When we look at the misclassifications in more detail, we notice that, especially short strokes, such as the dab and the short serve, are sensitive to the omission of the gyroscope. These strokes are barely recognized. The reason is that the angular speed is very different for these strokes, while the linear accelerations are quite similar. These problems are partly solved by adding the gyroscope data. The drive is then recognized with an accuracy of 62%. The global weighted average accuracy increases up to 89%. Rest moments and displacements are not really affected. It is clear to see that the overall results are more than ten percent higher by adding the extra sensor.

### 5.4. Influence of Sampling Frequency

We examined four different sampling frequencies: 12.5, 25, 50, and 100 Hz. As expected, we obtain the best results by using higher sampling frequencies. However, Both with and without gyroscope, we observe a fairly stable trend between 100 and 25 Hz. Here we suspect that these small differences are due to the randomization that occurs when training the model (e.g., dropout layers). At 12.5 Hz, we see a major drop in the accuracy of the predictions. As higher sample rates are more power consuming and require more memory/data communication, we conclude that 25 Hz is the recommended sampling frequency.

### 5.5. Position of Sensors

When we attach the sensor to the wrist, we notice that the model is confused between the dab and the netdrop. When we vary the weights in ensemble learning, no good values were found in order to distinguish these classes. Table 6 shows the results. Both strokes consist of a short forward movement followed by a short stroke movement. Placement of the sensor on the wrist makes it more difficult to properly distinguish these movements. The measurements where the sensor is placed on the upper arm do not provide better results than on the racket either.

Nevertheless, we still get very good results here. For the wrist, we obtain a weighted average accuracy of 93% at a sampling frequency of 100 Hz. On the upper arm, this is 96% as compared to the achieved 98% at the racket. Only the information measured on the wrist and upper arm shows less detail in the signal. The most detailed information was obtained by attaching the sensor to the bottom of the racket’s grip. The strokes on the two other positions were still distinguishable, but the moment of impact of the shuttle could not be detected. This information can be used to distinguish similar movements. The moment of impact is easy noticeable because of a very short high peak in the signal. These peaks can, for example, easily be found when analyzing the derivative of the signal. The difference between non-hitting and hitting the shuttle can be seen in Figure 6. Here, we show the same stroke twice where only during the second stroke the shuttle is hit. Knowing where this peak occurs during the movement can also help to further distinguish strokes or might be useful when analyzing the stroke movement. This position is also most user friendly, since the player does not have to attach sensors to the body. We conclude that we lose some information when we place the sensor further away from the racket, but these positions are still acceptable for activity recognition tasks. Additionally, we foresee that the support of left-handed players is straightforward as such strokes show similar trends with an inverted y-axis. In this case, the y-axis could be inverted before using the model, or the model could be further trained with using y-axis inverted data augmentation data and/or with data from left-handed players.

### 5.6. From Fine-Grained Actions to Stroke Types

Something that stands out, regardless of the position of the sensors or the sampling frequency, is the (often slight) confusion between similar strokes, such as the clear and the smash. Especially when we have data from multiple players of different levels, it is easy to understand that the model is prone to confusion. These strokes are very similar movements. The only difference is that the smash is likely to be a faster, shorter movement. Because the shuttle was not hit during the measurements of the data set, this peak cannot be seen in the data. The location of this peak could possibly clarify the classification between the two strokes. The timing for hitting the shuttle is slightly different to force the shuttle downwards for the smash or more flat or upwards for the clear. During a smash movement, the shuttle is hit a little later. If we would combine the clear and the smash in an overarching category, we see that the obtained accuracy is higher. When measuring at the wrist, the average accuracy is 91%. After combining the two classes, this becomes 97%. Another way to create supersets is by taking overhand and underhand strokes together or make a division based on offensive and defensive strokes. For example, Figure 7 shows the capabilities of our model to distinguish overhand and underhand strokes, resulting in even higher accuracies than for individual strokes.

### 5.7. Novel Weights-Based Neural Network Architecture

In the previous sections, we investigated the ability of our proposed model to distinguish scenarios of an unseen player. Accordingly, the model was trained on the data of one player and tested on the data of the other one. We notice a slight confusion between the clear and the drive on the one hand and the short serve and the netdrop on the other.

Although we get a large improvement by applying ensemble learning, we noticed, during our experiments, that the largest deviations occur when predicting the clear and the short serve (respectively, the longest and shortest stroke in time). When we make a prediction, we get a probability (Softmax) for all frames f per stroke type i. Per frame, the most likely class is then returned as a result. When we notice that there is confusion between two different classes and we know that one of the two is very likely to be incorrect (e.g., recognizing a short serve with a model based on long framesizes), we can correct this by using weights. These weights are determined according to the average duration of the movements. We increase the weights for a short service on the models with a smaller framesize to further improve the accuracy. Additionally, the weights for a clear on the models with a larger framesize obtain a higher value. As a result of these modifications, we obtain almost perfect results for the same scenario, as shown in Figure 8.

The formula below outlines our approach, where *predicted_classes* is the complete sequence of strokes recognized by the model in a scenario. For each stroke, the softmax-certainty, predict_f,i_, is multiplied by the corresponding weight_i_. The index that yields the highest value is the index of the resulting class.
$$predicted\_classes=\overset{F}{\underset{f=1}{⋂}}index_{predict_{f}}\left(max\left(weight_{i}\cdot predict_{f,i}\right)\right)$$

We conclude from these results that this approach, whereby activity classes that are better recognized for different frame durations, are given respectively higher weights, gives extremely good performance results. Similar approaches might also be useful for activity recognition in other sports or in completely different branches where signals need to be classified that vary in duration.

### 5.8. Low-Complexity Dnn versus Cnn

Finally, although we can achieve very good results by using a Convolutional Neural Network, we examine whether it is possible to achieve similar accuracy with a Deep Neural Network without convolutional layers. The main advantage of a DNN is the lower computational complexity, making it more suited for e.g., embedded implementations.

Table 7 shows the used layers and output dimensions. We again make use of the weighted ensemble learning technique. Similar to the more complex model, the DNN also has difficulty distinguishing similar movements, e.g., smash and clear. At a sampling frequency of 100 Hz, an accuracy of 93% is obtained (as compared to 98% for the CNN). The weighted average precision and recall also equal 93%. As such, this solution can be acceptable for low-cost implementations.

## 6. Future Work

An interesting extension of this research is the combination of this activity recognition with positioning. Because we attach great importance to a low cost and deployability in various environments, an Ultra-Wideband-based positioning system [28,29] could be a useful addition to the proposed sensor system. Accurately estimated positions can make activity recognition even more reliable, since not all strokes can be played on all positions on the court. For example, a netdrop is always played close to the net and never in the back of the court. It would also be of great added value to see which stroke was played at which location on the court. When we also track the relative positions between the players, we obtain a full description of the game situation. Therefore, the entire game can be fully reviewed and analyzed afterwards. This can be an interesting retrospect at high level where weaknesses in technique and tactics can be studied. It allows for more targeted coaching and better preparation for future matches.

## 7. Conclusions

In this paper, we have demonstrated that accurate fine-grained activity recognition of typical badminton strokes can be performed while using off-the-shelf sensors. We distinguish nine different activities: seven specific badminton strokes, displacements, and moments of rest. Accurate estimations can be made with a simple Convolutional Neural Network after fine-tuning the parameters. With only accelerometer data, we obtain a maximum precision of 86% for a CNN at a sampling frequency of 50 Hz. When accelerometer data and gyroscope data are combined, this value increases to 99%. Using a DNN, this value is about five percent lower. The model allows making a trade-off between accuracy and, for example, power consumption. Sample frequencies of more than 25 Hz appear to be sufficient to correctly classify the fast badminton movements. If we lower this frequency to 12.5 Hz, the precision of our CNN will drop to 77% when only accelerometer data are used, and to 89% when also a gyroscope are used. When only accelerometer data is used, it drops to 77%. When we merge the seven strokes into two disjoint supersets (i.e., overhand and underhand strokes), a weighted average classification accuracy of 99% is achieved.

To solve the problem of long and short movements, a novel weighted form of ensemble learning was used in this paper. The output of multiple models with different frame sizes were combined to achieve a reliable end result. The weights were determined based on the likelihood that the predicted class could be estimated while using the considered frame duration. This is not just a technique that is useful in our research, but it can be useful in any form of activity recognition. The best place to attach the sensors is at the bottom of the racket’s grip. Here, the smallest details in the stroke can be observed, making the movements easier to distinguish. The weighted accuracy for the wrist and upper arm is 93% and 96%, respectively. This is slightly lower than the accuracy of 98%, when the sensors are mounted on the racket.

## Figures and Tables

**Figure 1 sensors-20-04685-f001:**
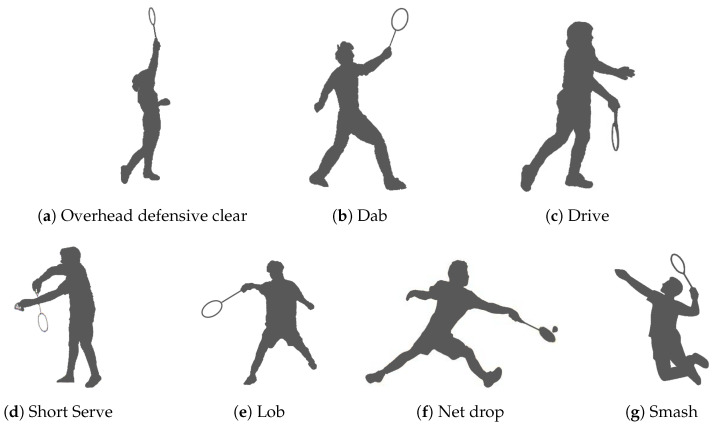
Classified strokes.

**Figure 2 sensors-20-04685-f002:**
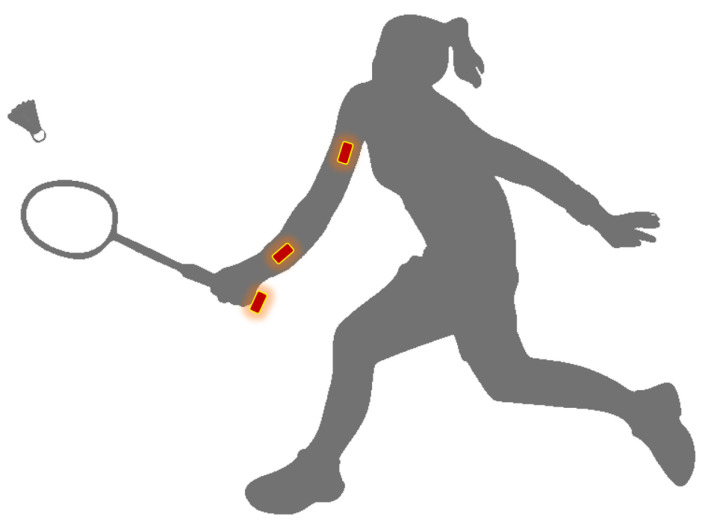
Positions sensors.

**Figure 3 sensors-20-04685-f003:**
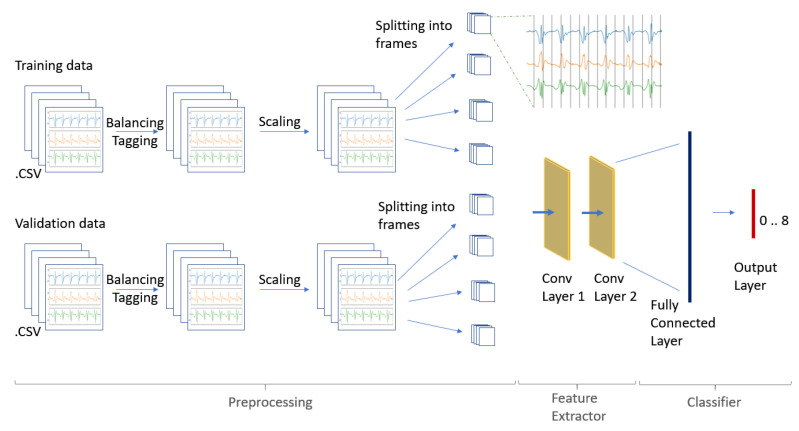
Overview preprocessing, feature extraction and classification.

**Figure 4 sensors-20-04685-f004:**
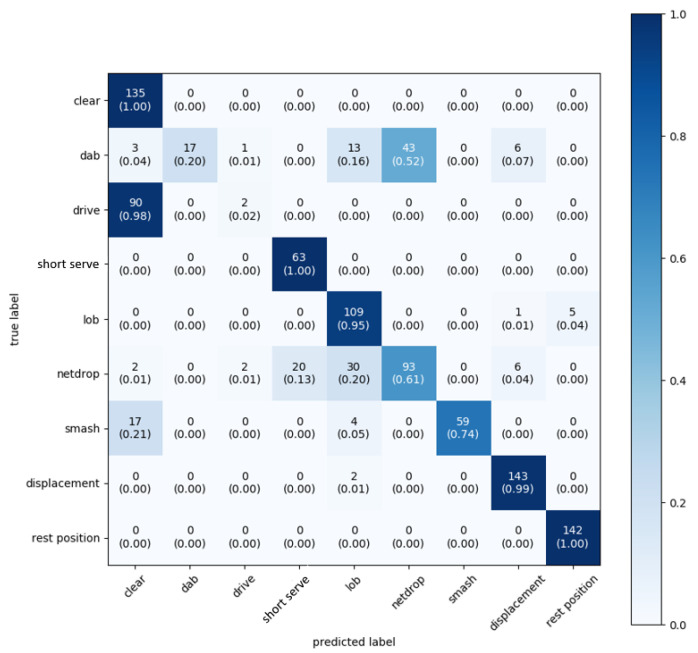
Confusion matrix: only accelerometer data and a sampling frequency of 12.5 Hz.

**Figure 5 sensors-20-04685-f005:**
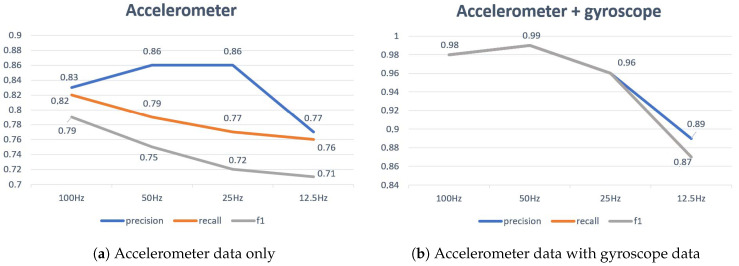
Precision, recall and f1 score measured with (**a**) accelerometer data only or in (**b**) combination with gyroscope data at sample frequencies of 100, 50, 25, and 12.5 Hz.

**Figure 6 sensors-20-04685-f006:**
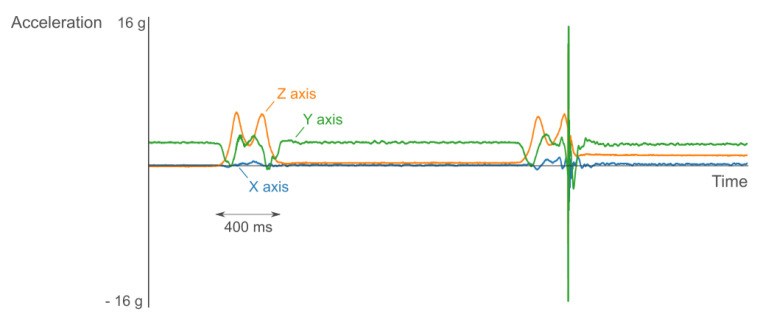
Impact shuttle when placing the sensor on the racket’s grip. The blue, green and orange line represent the respectively X, Y, and Z axis of the accelerometer. Left: stroke when not hitting the shuttle. Right: same stroke when the shuttle is hit. This shows a peak in the Y axis of the accelerometer when hitting the shuttle.

**Figure 7 sensors-20-04685-f007:**
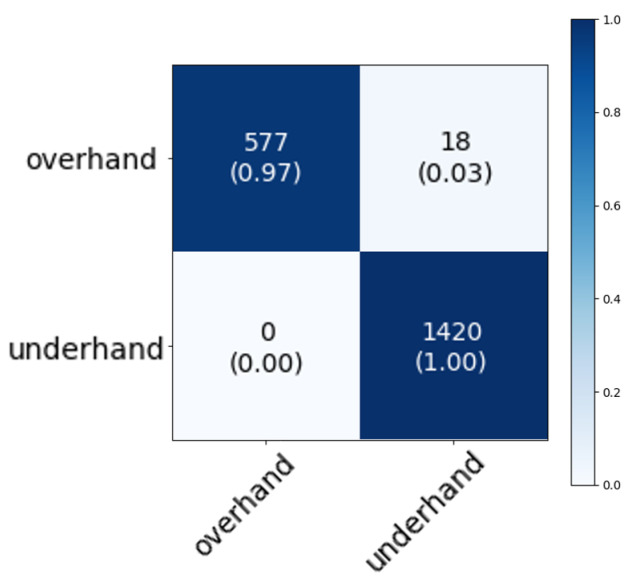
Confusion matrix for distinguishing overhand and underhand strokes. We notice that the overhand and underhand strokes are correctly classified with an average accuracy of 99%.

**Figure 8 sensors-20-04685-f008:**
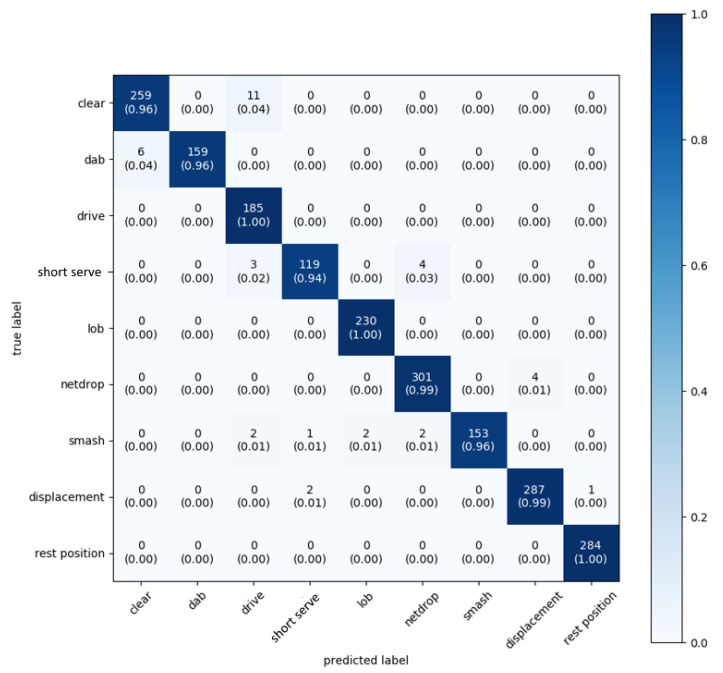
Novel weights-based neural network architecture. Confusion matrix: accelerometer and gyroscope data. A sampling frequency of 100 Hz and ensemble learning with extra weights are used. The average accuracy is 98%.

**Table 1 sensors-20-04685-t001:** Comparing recognized activities, number of distinguished classes, sensors, and methods proposed in this paper with related work. What makes our research unique is the optimization of every step within activity recognition.

Paper	Activity	Number of Classes	Sensors	Method/Positions	Machine Learning Approach
[15]	Daily activities	8	Accelerometer + gyroscope	Wrist, ankle	SVC, DNN
[15]	Coarse-grained sport categories	9	Accelerometer + gyroscope	Wrist, ankle	SVC, DNN
[17]	Tennis	5	Accelerometer + gyroscope	Wrist, waist	CNN
[8]	Badminton	5	Accelerometer + gyroscope	On the strings of the racket	KNN, SVM
[16]	Badminton	2	Cameras	Computer vision	SOFW
[18]	Badminton	2	Cameras	Computer vision	CNN
**This paper**	**Badminton**	**9**	**Accelerometer + gyroscope**	**Grip racket, wrist, upper arm**	**CNN, DNN**

**Table 2 sensors-20-04685-t002:** Layers CNN (frame size = 80).

Layer	Output Dimension
Input	80 × 6
Conv2D (16, 3 × 3), Relu	78 × 4 × 16
Batch normalization	78 × 4 × 16
Dropout 10%	78 × 4 × 16
Conv2D (32, 3 × 3), Relu	76 × 2 × 32
Dropout 20%	76 × 2 × 32
Flatten	4864
Dense (64), Relu, kernel_regularizer, bias_regularizer	64
Dense (9), Softmax	9

**Table 3 sensors-20-04685-t003:** Number of strokes in training and test data.

Stroke	# Training	# Test
Overhead Defensive Clear	150	31
Dab/Block	280	30
Drive	120	30
Short serve	65	31
Lob	168	30
Net drop	300	34
Smash	170	30

**Table 4 sensors-20-04685-t004:** Best frame size by stroke type.

Stroke	Frame Size (s)
Overhead Defensive Clear	1.60
Dab/Block	1.08
Drive	1.08
Short serve	0.68
Lob	1.28
Net drop	1.20
Smash	0.72

**Table 5 sensors-20-04685-t005:** Best accuracy by frame size (100 Hz, accelerometer + gyroscope).

Frame Size (s)	Accuracy (%)
0.40	80
0.60	81
0.80	88
1.00	88
1.20	92
1.40	88
1.60	93
**Ensemble learning**	**98**

**Table 6 sensors-20-04685-t006:** Most optimal f1 score after varying score weights in ensemble learning.

Position	Dab-F1 Score	Netdrop-F1 Score
Racket	0.99	0.98
Wrist	0.50	0.73

**Table 7 sensors-20-04685-t007:** Layers DNN (frame size = 80).

Layer	Output Dimension
Input	80 × 6
Dense (1024), Relu	80 × 1024
Dropout 20%	80 × 1024
Dense (512), Relu	80 × 512
Dropout 20%	80 × 512
Dense (256), Relu	80 × 256
Dropout 20%	80 × 256
Dense (128), Relu	80 × 128
Dropout 20%	80 × 128
Dense (32), Relu	80 × 32
Dropout 20%	80 × 32
Flatten	2560
Dense (9), Softmax	9

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
