# Peer review of "Badminton Activity Recognition Using Accelerometer Data"

_sensors, 2020, doi:10.3390/s20174685_

Round 1

Reviewer 1 Report

Dear Authors,

Thank you for the opportunity to review your paper. Below you will have my suggestions:

Introduction

While the flow of the text is right, the authors should do well to highlight the importance of their contribution and the fact that a study like this is not present in the literature. Also, even though badminton has not been analyzed using sensors and you certainly stated that in the related work section, there are other methods in the sports literature that have addressed this issue by other means (manual observation, video cameras…). Please, revise them to better contextualize your findings.

Data collection

During the two-hour record, how many actions were taken for the two athletes? It would be beneficial to have knowledge of the number of the 6 strokes recorded. You mentioned that 663954 samples were taken, but these are not actions. One can imagine that the number of actions would be 663954/100 = 6639 but the reader doesn’t really know.

Please remove “state-of-the-art” adjective when describing the sensors. It adds no information.

Regarding the sensors, please, add technical information of the models and remove “with high precision”: what is the precision of the model? Other technical values can be seen in the dataset, like, for instance, weight, a feature of high interest in this research.

“it must of course be converted ” → remove of course

There are many issues that shoud be addressed:

  • What do you do when signals go beyond dynamic range?
  • How do you tell between a signal that is intended to feed the network and the inherent noise of inertial devices? What is the criteria to discard this noise
  • Have you filtered the signals prior to feeding the neural network?
  • You mentioned downsampling to 25Hz or 50Hz. When do you apply such processing? What is the criteria?

“class is perfect for our goal.” → substitute perfect

Results.

Please, provide the number of actions fed to the model, as you do with the prediction (Table 3)

In order to give a value of accuracy, I assume you knew in advance the exact actions before putting them into the system to be classified. How did you obtain such information?

There is something not clear: you said “Further increasing the sampling rate to 50 and 100 Hz produces similar results.”,, which I think is 96%, yet you also claim “at 100 Hz the drives are recognized 48% of the time using only accelerometer data.” Can you explain this please?

Can you please clarify the difference between sample frequency and sampling rate? It seems there is some confusion since in 5.4 Influence of sampling rate, you claimed “by using higher sampling frequencies”

Please define f1 score.

Reviewer 2 Report

The authors present a study, which deals with a classification of a badminton activity using accelerometer data and Convolutional Neural Networks (CNN)/Deep Neural Networks (DNN). Seven typical strokes were selected for classification. The authors have used the data from one player for training and the data from another player for testing the classifier. The impact of sampling rate, type of data (accelerometer data v. accelerometer and gyroscope data) and type and configuration of the classifier have been evaluated in the study.

The paper is well prepared and organized, but the study shows some imperfections which should be clarified before accepting the paper.

Major Comments:

  1. Only two players have been involved in the study, one person for training the classifier, and the other one for the evaluation of the classification. Both of them are right-handed players. It is not clear from the study if using only two players is sufficient for reliable evaluation. What impact on the results would using more players have on the evaluation of the classification? How would the results be influenced if one of the players will be a left-handed player? It is evident that all players are not right-handed in reality.
  2. The authors have used the strokes without hitting the shuttle. It seems a little bit odd and should be clarified because the real strokes hit the shuttle. Moreover, Fig. 6 proves that the signals obtained while hitting the shuttle are different from the signals measured without hitting the shuttle.
  3. The important phase of the classification of continuous data is data segmentation. Please, explain in detail how are the data segmented before the classification.
  4. The goal of the research is not understandable from the paper. How could the results be used during the game or badminton training? How can the players or trainees apply the proposed methods?

Minor Comments:

  1. The authors have selected seven important strokes for classification (section 3.2), which should be shown in Fig. 1, as they stated on page 4. Surprisingly, Fig. 1 shows only 6 strokes. It is confusing and should be clarified.
  2. 6 has no axes. Please, complete the figure at least with values and units.

Reviewer 3 Report

In this manuscript, the authors have used wearable sensors to capture badminton players' movements. Then they have proposed a convolutional neural network to classify nine activities. Also, they have verified the sampling rates and the position of the sensors. In my opinion, the paper is well-written and has enough quality to be published in Sensors. However, I have the following comments to improve the quality of the article:

1) Please, inform the weight of the attached sensors to the players and the rackets.

2) In Subsection 3.2, you have defined seven strokes, but only six of them are depicted in Figure 1. Please, show the Overhead Defensive Clear too.

3) Please, describe the computational burden of the proposed method, such as the processing time, etc.

4) In the last paragraph of the Introduction, when the organization of the paper is explained, you have referred the sections by the word "Chapter", like Chapter2, Chapter 3, etc. Generally, a paper is divided into sections rather than chapters. Please, utilize the word "Section".

5) There is a typo at the caption of Figure 4: accelerometerdata --> accelerometer data (please, add the space between these words).

Round 2

Reviewer 2 Report

The authors have updated the paper and made some improvements which increased the quality of the paper. Thanks for these improvements.

As I can judge, the authors have reflected all comments. I have an only minor comment yet.

Minor Comments:

1) to cover letter, reviewer 2, comment 1:

I agree with the presented clarification. Please, mention the left-handed players' problem briefly in the text. I think it is an important matter.

Author Response

Thank you for your comment, we have added the clarification in the paper under subsection 5.5 Position of sensors: Lines 314-317: Additionally, we foresee that the support of left-handed players is straightforward as such strokes show similar trends with an inverted y-axis. In this case, the y-axis could be inverted before using the model, or the model could be further trained with using y-axis inverted data augmentation data and/or with data from left-handed players.
